# Conformational change of Dishevelled plays a key regulatory role in the Wnt signaling pathways

Ho-Jin Lee[1], De-Li Shi[2,3]*, Jie J Zheng[1,4]*

[1]Department of Structural Biology, St. Jude Children's Research Hospital, Memphis, United States; [2]Laboratoire de biologie du développement, Institut de Biologie Paris-Seine, Sorbonne Universités, Paris, France; [3]School of Life Sciences, Shandong University, Jinan, China; [4]Departments of Ophthalmology, David Geffen School of Medicine at University of California, Los Angeles, Los Angeles, United States

**Abstract** The intracellular signaling molecule Dishevelled (Dvl) mediates canonical and non-canonical Wnt signaling via its PDZ domain. Different pathways diverge at this point by a mechanism that remains unclear. Here we show that the peptide-binding pocket of the Dvl PDZ domain can be occupied by Dvl's own highly conserved C-terminus, inducing a closed conformation. In *Xenopus*, Wnt-regulated convergent extension (CE) is readily affected by Dvl mutants unable to form the closed conformation than by wild-type Dvl. We also demonstrate that while Dvl cooperates with other Wnt pathway elements to activate canonical Wnt signaling, the open conformation of Dvl more effectively activates Jun N-terminal kinase (JNK). These results suggest that together with other players in the Wnt signaling pathway, the conformational change of Dvl regulates Wnt stimulated JNK activity in the non-canonical Wnt signaling.

*For correspondence: de-li.shi@ upmc.fr (D-LS); jzheng@jsei.ucla. edu (JJZ)

**Competing interests:** The authors declare that no competing interests exist.

## Introduction

The multiple Wnt signaling–related pathways are crucial to various developmental processes (*Logan and Nusse, 2004*; *Angers and Moon, 2009*). By regulating the cellular β-catenin level, canonical Wnt signaling controls cell fate, while non-canonical Wnt signaling plays a key role in controlling convergent extension (CE) and polarized cellular orientation. Dishevelled (Dvl, or Dsh in *Drosophila*), a key component of both Wnt signaling pathways, relays Wnt signals downstream from the membrane-bound Wnt receptor Frizzled (Fz) (*Noordermeer et al., 1994*; *Theisen et al., 1994*; *Axelrod et al., 1998*; *Wong et al., 2003*; *Park et al., 2005*; *Wang et al., 2006*; *Schwarz-Romond et al., 2007*; *Simons et al., 2009*). While Dvl mediates both canonical and non-canonical Wnt signals, different pathways diverge at this point. Therefore, Dvl has been described as the 'policeman' at the intersection who directs different Wnt signals in different directions (*Boutros and Mlodzik, 1999*). However, the mechanism by which Dvl relays Wnt signals from Fz to different downstream components is not well understood.

Dvl contains highly conserved DIX, PDZ, and DEP domains and a highly conserved extreme C-terminus (*Figure 1*) (*Wharton, 2003*; *Wallingford and Habas, 2005*). The PDZ and DIX domains are reported to be involved in canonical Wnt signaling, while the PDZ and DEP domains play a crucial role in non-canonical Wnt signaling (*Sokol et al., 1995*; *Axelrod et al., 1998*; *Boutros et al., 1998*; *Li et al., 1999*; *Moriguchi et al., 1999*; *Yamanaka et al., 2002*; *Gao et al., 2010*). The central PDZ domain not only participates in both pathways but also binds directly to the membrane-bound Wnt receptor Fz (*Wong et al., 2003*). Many Wnt signaling regulators have been reported to mediate the different Wnt signaling pathways by interacting directly with the PDZ domain of Dvl

**eLife digest** The development of an animal embryo depends on a number of signaling pathways that pass information from the outside of the cell to the inside. These pathways include Wnt signaling, which also regulates cell growth. The pathways must be precisely controlled; abnormal Wnt activity has been implicated in several human diseases, ranging from heart disease to cancer.

Wnt signaling is complex, and actually comprises two major pathways: the canonical pathway (which depends on a protein called β-catenin) and the PCP pathway (which doesn't depend on β-catenin). Both pathways are triggered when Wnt molecules bind to receptors on the outside of the cell. These receptors pass the signal into the cell and to a protein called 'Dishevelled' (or 'Dvl' for short). This protein then passes the signal on through either the canonical or PCP pathway. Nevertheless it is not clear how the Dishevelled protein can direct the signal specifically down either one of these pathways.

Lee et al. now show that the Dishevelled protein can take on at least two different shapes. When it is 'closed', one end of the protein is tucked inside a pocket elsewhere on the protein's surface. But when Dishevelled is 'open', this end of the protein moves out of this pocket. Further experiments using frogs (called *Xenopus*, which are commonly used in research) reveal that mutant versions of Dishevelled that were unable to take on the closed form strongly affected an aspect of the frog's development that involves the PCP pathway.

Lee et al. then demonstrate that while Dishevelled cooperates with several other Wnt pathway components to activate the canonical pathway, the open form of Dishevelled activates the PCP pathway. The next challenge following on from this work is to find out how Wnt molecules binding to the receptor trigger the shape change in Dishevelled.

(*Wharton, 2003*; *Wallingford and Habas, 2005*; *Gao and Chen, 2009*). Notably, in most species the extreme C-terminus of Dvl resembles a class III PDZ-binding motif (E/D-X-Φ, where Φ represents hydrophobic residues such as F, I, L, M, or V), while that of Dsh resembles a class II PDZ-binding motif (Φ-X-Φ) (*Figure 1*) (*Tonikian et al., 2008*; *Lee and Zheng, 2010*). Because these two motifs suggest the possibility of intramolecular binding, we hypothesized that the C-terminus of Dvl/Dsh binds intrinsically to the Dvl PDZ domain.

Here we use biophysical methods to investigate the interaction of the Dvl extreme C-terminus with the Dvl PDZ domain and demonstrate that Dvl adopts a closed conformation. We also show, in a *Xenopus* model, that disruption of this intramolecular interaction activates Jun N-terminal kinase (JNK) and enhances the CE phenotype associated with activation of non-canonical Wnt signaling. Further, we demonstrate that a Dvl PDZ-binding peptide or small molecule that inhibits canonical β-catenin signaling enhances JNK activity by releasing the Dvl C-terminus from its autoinhibitory closed conformation.

## Results

### The Dvl C-terminus and PDZ domain directly interact

After our initial pull-down test indicated that Dvl C-terminus might interact with the Dvl PDZ domain, we decided to use two different biophysical assays to determine the binding affinity of the Dvl PDZ domain for the Dvl-C peptide and for a peptide derived from the C-terminus of *Drosophila* Dsh ('Dsh-C peptide'). We first used a competitive binding assay. Binding of the PDZ domain to a fluorescently-labeled peptide (Rox-DprC) derived from the C-terminus of Dapper (Dpr), a known binding partner of the Dvl PDZ domain (*Cheyette et al., 2002*), was monitored by fluorescence polarization. Both Dvl-C and Dsh-C peptides competitively inhibited the interaction of Rox-DprC with the Dvl-1 PDZ domain (*Figure 2*), indicating that the three peptides competed for the same binding site on the PDZ domain. The inhibition constants ($K_I$) calculated from two independent experiments ($12.3 \pm 7.8$ μM for the Dvl-C peptide and $26.8 \pm 8.4$ μM for the Dsh-C peptide) were similar to the binding affinity of Rox-DprC to the PDZ domain ($K_D \sim 7.9 \pm 0.9$ μM) (*Lee et al., 2009a, 2009b*). We then measured the binding between Dvl-C and the Dvl-1 PDZ domain by isothermal titration calorimetry (ITC) (*Figure 3*) and obtained a $K_D$ of $7.0 \pm 0.7$ μM, which is consistent with the $K_I$ value observed in the fluorescence study.

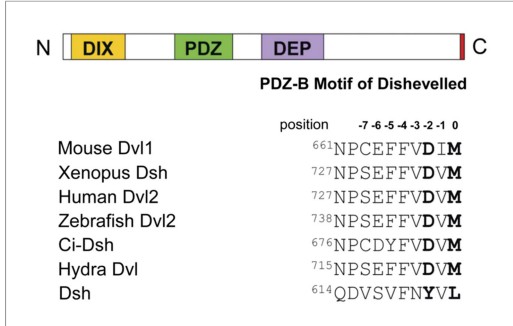

**Figure 1**. The C-terminal tail of Dishevelled (Dvl) is a PDZ domain binding motif. Sequence alignment of the C-terminus of Dvl/Dsh from selected species (*Wallingford and Habas, 2005*), showing residue numbers.

## Structural analysis of the Dvl PDZ domain in complex with the Dvl C-terminus

We next conducted NMR chemical shift perturbation studies (*Wong et al., 2003*). Unlabeled Dvl-C and Dsh-C peptides were repetitively titrated into a solution of $^{15}$N-labeled Dvl-1 PDZ domain. During these titrations, the NMR resonance of the residues in the PDZ domain's peptide binding site (*Wong et al., 2003*; *Lee et al., 2009a*, *2009b*) exhibited large chemical shift perturbations (*Figure 4*), further suggesting that both peptides interact with the Dvl PDZ domain. Moreover, during the Dvl-C peptide titration, a few resonances of the PDZ domain disappeared and reappeared (*Figure 4*), indicating that the Dvl-C peptide binds to the Dvl1 PDZ domain with higher binding affinity than the Dsh-C peptide does and that the complex is formed in the intermediate exchange range on the NMR time scale.

To structurally analyze the binding of the Dvl-1 PDZ domain and the Dvl-C peptide, we determined the solution structure of the complex that formed (*Figure 5*). The Dvl PDZ domain contains six β-strands (βA~βF) and two α-helix (αA and αB) structures (*Wong et al., 2003*; *Lee et al., 2009a*, *2009b*). As expected, we found that the Dvl-C peptide fits into the αB/βB peptide-binding groove of the Dvl PDZ domain and forms an additional β-strand with the αB-structure of the Dvl PDZ domain (*Figure 5A–C*). Nuclear Overhauser effect (NOE) data indicated that six residues in the Dvl-C peptide participate in binding (*Figure 5—source data 1, 2*). The side chain of Met(0) in the Dvl-C peptide is located within a hydrophobic pocket formed by the side chains of residues Leu262, Ile264, and Ile266 in the βB-structure and residues Val325, Leu321, and Val318 in the αB-helix structure (*Figure 5C,D*). Notably, the side chain of residue Asp(-2) in the Dvl-C peptide forms a hydrogen bond with the side chain of residue Arg322 in the Dvl PDZ domain (*Figure 5E*). Consistent with this finding, the Arg322Ala substitution dramatically weakened the binding of Dvl PDZ to the Dvl-C peptide (*Figure 5—figure supplement 1*) (*Lee et al., 2009a*). The side chains of two hydrophobic Dvl-C residues, Val(-3) and Phe(-5), interact with the side chains of the Val318 and Ile266 residues in the Dvl PDZ domain (*Figure 5E*).

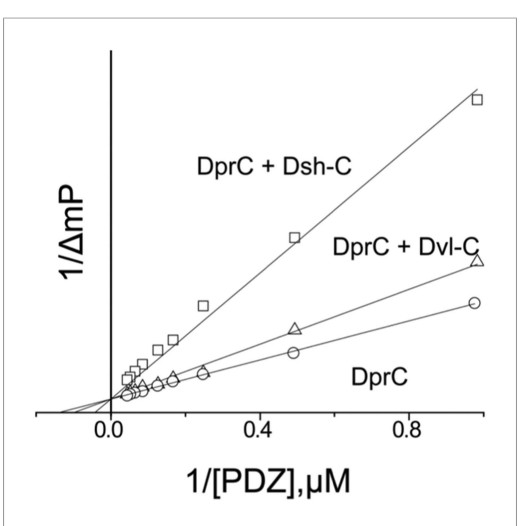

**Figure 2**. Competitive binding experiments. The $K_D$ value of the fluorescence-labeled Dapper (Dpr)-derived peptide Rox-DprC was obtained by plotting $1/\Delta mP$ vs $1/[PDZ]$, where $\Delta mP$ is the fluorescence polarization change (×1000) of Rox-DprC and [PDZ] is the concentration of the PDZ domain of Dvl. The $K_I$ values of the Dvl-C and Dsh-C peptides were obtained by using the equation $K_D^{app} = (K_D/(1 + [I]/K_I))$.

## The binding pocket of Dvl PDZ is occupied by its intrinsic C-terminus

To verify that the binding pocket of the Dvl PDZ domain is occupied by the intrinsic C-terminus, we generated two constructs: mC1 (residues 251–695 of mouse Dvl-1), containing the PDZ and DEP domains (with an intact PDZ-binding motif) and mC1-CΔ7 (residues 251–688), lacking the PDZ-binding motif. We then examined the binding of Rox-DprC to the two constructs. Little polarization change was observed when mC1 proteins were added to Rox-DprC solution, whereas large polarization changes were induced

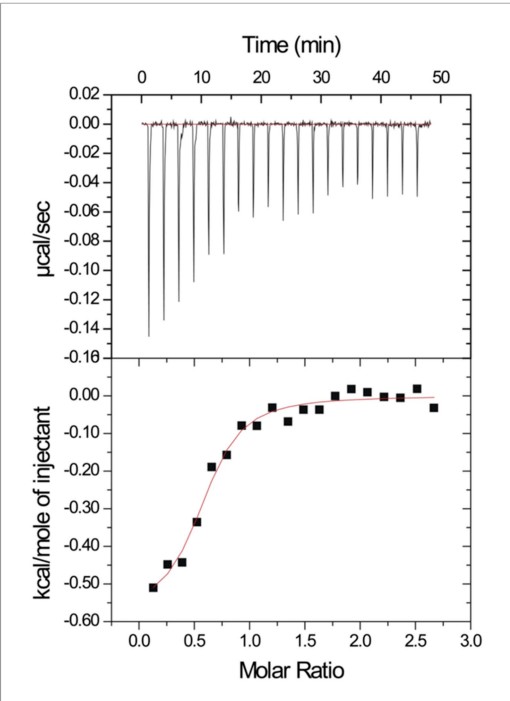

**Figure 3**. Isothermal titration calorimetry experiment. The MicroCal Auto-iTC-200 was used to obtain the binding affinity of Dvl-C peptide and Dvl PDZ protein in 50 mM phosphate buffer. The concentration of Dvl-C peptide in the syringe was 1.05 mM and the concentration of Dvl PDZ domain in the cell was 0.114 mM. The $K_D$ value was averaged from two independent experiments at 25°C.

by adding mC1-CΔ7 (*Figure 6*), indicating that the binding pocket of the Dvl PDZ domain was occupied by its intrinsic C-terminus in the mC1 construct. The binding affinity between Rox-DprC and mC1-CΔ7 was 3.7 ± 0.5 μM, closely approximating the affinity between Rox-DprC and the Dvl-1 PDZ domain. In contrast, the estimated binding affinity between Rox-DrpC and mD1 was greater than 70 μM, suggesting not only that the Dvl PDZ domain binds intrinsically to the Dvl C-terminus but also that this binding interferes with interactions between the Dvl PDZ domain and its other binding partners in the Wnt signaling pathways.

## Dvl conformational change significantly affects Wnt-regulated cell polarity

After establishing that Dvl can adopt a 'closed' conformation in which its C-terminus binds to its PDZ domain, we used a *Xenopus* model (*Xenopus* Dvl-2, XDsh) (*Sokol et al., 1995*; *Sokol, 1996*; *Umbhauer et al., 2000*) to investigate how this intrinsic interaction affects the role(s) of Dvl in the Wnt signaling pathways. We first generated two constructs, both myc-tagged at the N-terminus: wild-type *XDsh* (residues 1–732) (*Sokol, 1996*) and mutant *XDsh-CΔ8* (residues 1–724, lacking the C-terminal PDZ-binding motif and therefore unable to form the intrinsic interaction between the its PDZ domain and the C-terminus). To avoid interfering the function of Dvl C-terminus,

we placed the myc-tag at the N-terminus. However, the tag may affect the N-terminal DIX domain or potentially have some nonspecific effects. Therefore, in the *Xenopus* studies, we always paired the two constructs in the experiments to minimalize the potential of unexpected effects.

To examine canonical Wnt signaling, we used a luciferase assay with a *siamois* promoter reporter (*Brannon et al., 1997*). The *Xenopus* Wnt target gene construct with *siamois* promoter-driven luciferase reporter and equivalent quantities of *XDsh-CΔ8* and wild-type *XDsh* mRNA, respectively, were coinjected into the animal pole region of 2-cell *Xenopus* embryos, and luciferase activity was then assayed in ectodermal explants dissected at the late blastula stage. Three different doses (80 pg, 200 pg, and 500 pg) of *XDsh* and *XDsh-CΔ8* mRNAs were used. At low and intermediate doses, neither XDsh nor XDsh-CΔ8 substantially affected canonical Wnt signaling (not shown). At the high dose, both constructs enhanced canonical Wnt signaling, but wild-type XDsh was a stronger activator (*Figure 7*), consistent with a previous report showing involvement of the extreme C-terminal region of Dvl in activation of canonical Wnt signaling (*Tauriello et al., 2012*).

To examine how the conformational change of Dvl affects the Wnt-regulated CE phenotype (associated with the non-canonical Wnt signaling), we injected three different doses of wild-type *XDsh* and of *XDsh-CΔ8* mRNA into the dorsal blastomeres of 4-cell *Xenopus* embryos. At all three doses (80 pg, 200 pg, and 500 pg), *XDsh-CΔ8* mRNA caused greater body axis shortening and dorsal tail flexion than did wild-type *XDsh* (*Figure 8A*), suggesting that XDsh-CΔ8 is more active than wild-type XDsh in inducing the CE phenotype. We then compared Xdsh-CΔ8 with another Dvl mutant that lacks the PDZ domain, the well-established Xdd1 (*Sokol, 1996*). Like XDsh-CΔ8, Xdd1 is in 'open' conformation because it has no PDZ domain to interact with its C-terminus. We injected equal quantities (500 pg) of N-terminal myc-tagged *Xdd1*, *XDsh-CΔ8*, and wild-type *XDsh* mRNA, respectively, into the dorsal blastomeres of 4-cell *Xenopus* embryos and found that the phenotypes

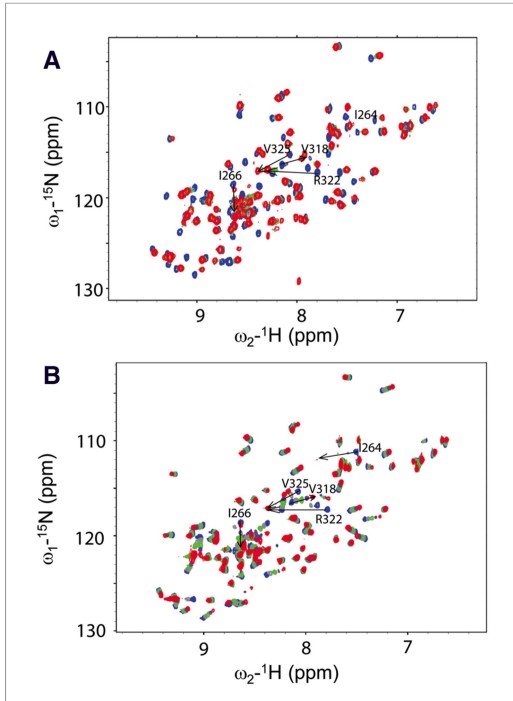

**Figure 4**. Direct interaction of the Dvl C-terminus and PDZ domain. (**A**) Overlap of $^{1}$H-$^{15}$N HSQC spectra of $^{15}$N-labeled PDZ domain without (blue) and with the unlabeled peptide (SEFFVDVM) derived from the extreme C-terminus of Dvl. Free: blue; final: red; ratio of peptide:protein = 20:1. (**B**) Overlap of $^{1}$H-$^{15}$N HSQC spectra of $^{15}$N-labeled PDZ domain without (blue) and with unlabeled peptide (QDVSVSNYVL) derived from the C-terminus of Drosophila Dsh (Dsh-C). Blue: free; red: final; ratio of peptide:protein = 20:1.

were very similar after injection of *XDsh-CΔ8* and after injection of *Xdd1* (*Figure 8B*), suggesting that the 'open' Dvl is more active than the 'closed' wild-type Dvl in inducing the *Xenopus* CE phenotype.

## The open conformation of Dvl activates JNK more effectively

In *Xenopus* embryos, JNK is known to regulate CE through non-canonical Wnt-PCP signaling (*Sokol, 1996*, *2000*; *Jones and Chen, 2007*), and either hyperactivation or JNK depletion dysregulates CE (*Yamanaka et al., 2002*). We therefore investigated how the change of Dvl conformation affects JNK activity. We overexpressed wild-type XDsh, Xdd1, and XDsh-CΔ8 in the ventral regions of 4-cell *Xenopus* embryos and dissected the ventral regions at the early gastrula stage to examine JNK phosphorylation by western blot. Overexpression of wild-type XDsh induced slightly greater JNK phosphorylation than observed in uninjected control cells. However, both Xdd1 and XDsh-CΔ8 induced JNK phosphorylation more potently (*Figure 9A*), suggesting that deletion of the XDsh C-terminal region renders XDsh more active in the non-canonical Wnt pathway. This finding is consistent with an early report of more effective JNK activation in COS-7 cells expressing mutant Dvl lacking the PDZ domain than in cells expressing wild-type Dvl (*Sokol et al., 1995*; *Axelrod et al., 1998*; *Li et al., 1999*; *Moriguchi et al., 1999*; *Yamanaka et al., 2002*; *Gao et al., 2010*).

To further demonstrate that the conformation of Dvl regulates its activation of JNK, we used the AP1-luciferase reporter (*Rui et al., 2007*) to monitor JNK activation in whole *Xenopus* embryos. When equal quantities (500 pg) of *XDsh-CΔ8* or wild-type *XDsh* mRNA were injected into the dorsal region of 4-cell *Xenopus* embryos and luciferase activity was assayed at the late gastrula stage, XDsh-CΔ8 clearly activated AP1-luciferase activity more strongly (*Figure 9B*), indicating that Dvl in the open conformation activates JNK more potently than Dvl in the closed conformation.

To clarify the role of the Dvl C-terminus in JNK activation, we analyzed activin-induced changes in the length and shape of *Xenopus* ectodermal explants; these changes represent the CE phenotype associated with non-canonical Wnt signaling (*Figure 9C–H*). In *Xenopus*, both activation and inhibition of non-canonical Wnt signaling result in planar cell polarity (PCP) defects and produce CE phenotype (*Djiane et al., 2000*). Ectodermal explants from control embryos and embryos injected with different mRNAs that encode Dvl and different mutants were dissected at the early blastula stage and treated with activin. The phenotypes were monitored at equivalent early neurula stages. As expected, *XDsh-CΔ8* and the dominant-negative mutant *Xdd1* strongly inhibited the activin-induced elongation of ectodermal explants (CE outcomes) as compared to controls, while wild-type *XDsh* had little effect (*Figure 9D–F*). However, co-expression of a dominant-negative JNK mutant (dnJNK), which inhibits non-canonical Wnt signaling (*Yamanaka et al., 2002*; *Carron et al., 2005*), with *XDsh-CΔ8* or *Xdd1* efficiently rescued explant elongation (*Figure 9G,H*), indicating that the CE defects induced by XDsh-CΔ8 and Xdd1 are the result of JNK activation. In addition, we also confirmed that co-expression of dnJNK with XDsh could also rescue explant elongation (not shown).

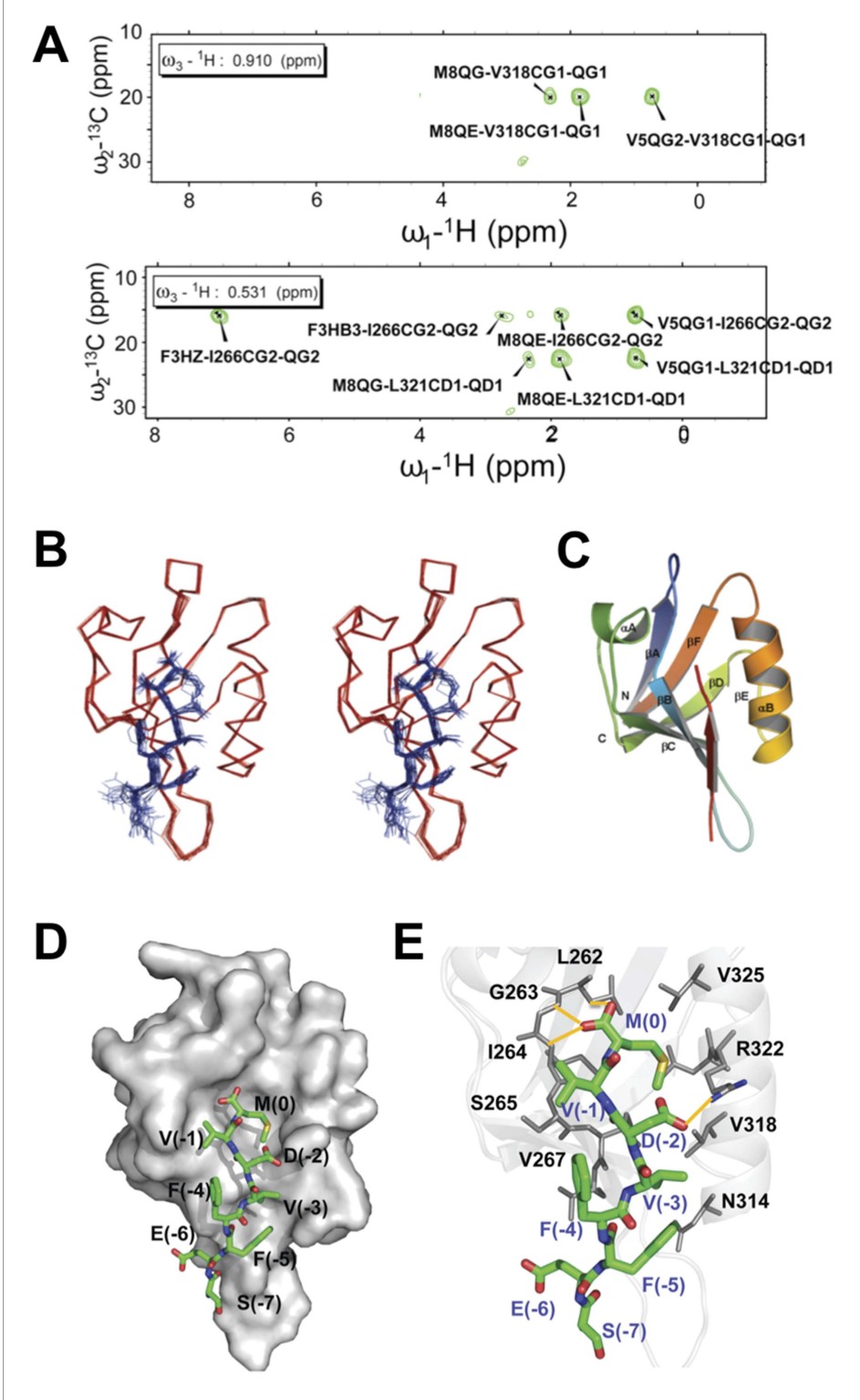

**Figure 5**. Solution NMR structure of Dvl PDZ domain in complex with Dvl-C peptide. (**A**) 2D plane of 3D $^{13}$C-F1-half-filtered F2-edited NOESY-HSQC spectrum (mixing time, 300 ms) at 15°C. The ratio of peptide: protein was 10:1. [$^{13}$C,$^{15}$N-PDZ] = ~1 mM. (**B**) A stereo view of the backbone of 15 superimposed structures of the Dvl PDZ–Dvl-C peptide complex. (**C**) Ribbon diagram of the lowest-energy structure of the Dvl PDZ/Dvl-C peptide complex. (**D**) Surface of Dvl-1 PDZ bound to Dvl-C peptide (carbon, green; nitrogen, blue; sulfur, yellow; oxygen, red; hydrogen atoms are omitted for clarity). (**E**) Structural details of the Dvl-C peptide–PDZ

*Figure 5. continued on next page*

*Figure 5. Continued*

domain complex. The side chain of Asp(-2) in the Dvl-C peptide forms a hydrogen bond with the side chain of Arg322 on the αB-structure.

The following source data and figure supplement are available for figure 5:

**Source data 1**. Intermolecular NOEs between the Dvl-C peptide and the PDZ domain obtained from [13]C-half-filtered NOESY-HSQC spectra[a].

**Source data 2**. Structure statistics for the 15 lowest-energy peptide-PDZ complexes.

**Figure supplement 1**. The mutant Dvl-1 PDZ (R322A) domain binds more weakly than wild-type Dvl-1 PDZ domain to the Dvl-C peptide.

Finally, to further confirm that the effect of Dvl's open conformation on the CE phenotype resulted from JNK activation, we coinjected *dnJNK* with *XDsh-CΔ8* or *Xdd1*, and analyzed the phenotypes in *Xenopus* whole embryo. Injection of *XDsh-CΔ8* or *Xdd1* alone resulted in embryos with short and bent axis, reflecting CE defects. However, the abnormal CE phenotype was substantially rescued when the embryos were coinjected with *dnJNK* (*Figure 9I*), suggesting that JNK activation induced by the Dvl open conformation (i.e., use of XDsh-CΔ8 or Xdd1) had resulted in the CE phenotype.

## Obstruction of the Dvl PDZ domain activates JNK

Many Wnt signaling regulators directly bind to and inhibit the Dvl PDZ domain (*Wharton, 2003*; *Wallingford and Habas, 2005*). To block Wnt signaling transduction at the Dvl level, we also developed a series of small-molecule inhibitors that disrupt Fz–Dvl interaction (*Shan et al., 2005*, *2012*; *Grandy et al., 2009*; *Shan and Zheng, 2009*; *Lee et al., 2009b*). Because all of the Wnt-regulating proteins and small molecule inhibitors target the site on the surface of the Dvl PDZ domain that binds to the molecule's own C-terminus, these agents should release the Dvl C-terminus from its intramolecular binding and open the closed conformation of Dvl. These molecules can be used to probe the effect of the conformational change of Dvl. For this study we chose a small molecule inhibitor of Fz–Dvl interaction, compound 3289–8625 (*Grandy et al., 2009*) and a protein inhibitor of Wnt signaling, TMEM88 (*Lee et al., 2010*).

We previously showed that the small molecule 3289–8625 penetrates the *Xenopus* embryo and binds to the PDZ domain of Dvl (*Grandy et al., 2009*). Microinjection of embryos with *XDsh* mRNA and incubation in medium containing compound 3289–8625 increased the prevalence of the non-canonical Wnt signaling–associated CE phenotype (*Figure 10A*). We also previously reported the protein TMEM88 to be a novel Wnt signaling inhibitor whose C-terminal region binds to the Dvl PDZ domain (*Lee et al., 2010*). When we coinjected embryos with mRNAs encoding the C-terminal half of TMEM88 (TMEM88-C) and wild-type XDsh, abnormal CE phenotypes were again more prevalent than in embryos injected only with *XDsh* mRNA (*Figure 10A*); this result is consistent with the fact that gain-of-function of PCP signaling induced by activated Dvl affects gastrulation movements and disrupts axis elongation (*Djiane et al., 2000*). CE defects were much less prevalent in control embryos incubated in 3289–8625 or injected with TMEM88-C mRNA alone (*Figure 10A*).

To further dissect how inhibition of the Dvl PDZ domain affects Dvl's role in the Wnt/β-catenin and Wnt/JNK signaling pathways, we used the TOPFLASH and AP1-luciferase reporter assays. To target the Dvl PDZ domain with TMEM88-C, we again coinjected the embryos with mRNAs encoding TMEM88-C and wild-type XDsh. As we reported previously (*Lee et al., 2010*), binding of the PDZ domain by TMEM88-C antagonized Wnt/β-catenin activity induced by Dvl overexpression (*Figure 10B*). Further, this opening of the Dvl conformation potentiated Wnt/JNK signaling induced by Dvl overexpression (*Figure 10C*).

## Discussion

Here we showed that a C-terminal motif of Dvl can bind intrinsically to the Dvl PDZ domain, forming a 'closed' conformation. Although the binding affinity of this reaction is not notably

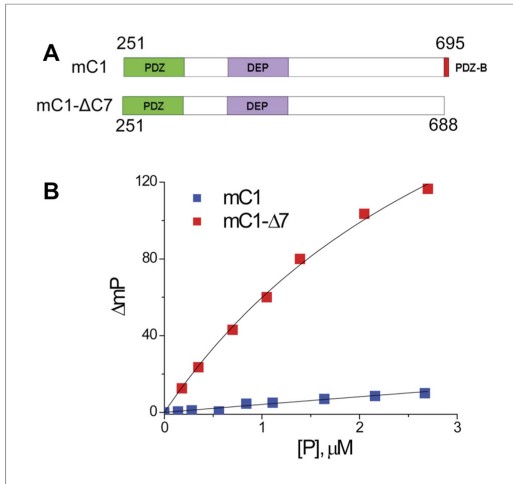

**Figure 6**. The binding pocket of the Dvl PDZ domain can be occupied by its intrinsic C-terminus. (**A**) Schematic representation of protein constructs mC1 (residues 251–695) and mC1-CΔ7 (residues 251–688) numbered according to the mouse Dvl-1 protein sequence. (**B**) Polarization change of the fluorescence-labeled peptide Rox-DprC (Rox-SGSLKLMTTV, derived from the C-terminus of Dpr) after addition of mC1-CΔ7 and mC1 proteins in 50 mM phosphate with 0.3 M NaCl and 6 mM β-mercaptoethanol. For the binding of Rox-DprC to mC1, KD is 3.8 ± 0.5 μM the value was obtained by fitting the titration data with the equation: $\Delta mP = \Delta mP_{max} \times [P]/([P] + K_D)$, where $\Delta mP$ is the polarization change of Rox-DprC, [P] is the concentration of protein, and both $K_D$ and $\Delta mP_{max}$ are the fitting variables. For the binding of Rox-DprC to mC1-CΔ7, $K_D$ was estimated as 68 ± 5 μM. Because of the limitation in the titration study, to estimate the $K_D$ value, although we used the same equation to fit the titration data, in the fitting Kd was the only variable and the maximum polarization change, $\Delta mP_{max}$, was fixed to the value that was obtained in the titration study of Rox-DprC binds to mC1-CΔ7.

high, the closed conformation is stable and may predominate among intracellular Dvl proteins (*Zhou et al., 2006*). To compare the effects of the two Dvl conformations, we examined a C-terminal–truncated Dvl mutant, the established Xdd1 mutant (neither of these forms the 'closed' conformation in solution), and wild-type Dvl in different *Xenopus* assays and found that both of the 'open' Dvl constructs significantly enhanced the CE phenotype mediated by Wnt-JNK signaling. To further support this observation we observed the competitive binding to the PDZ domain of the Dvl C-terminus and two agents known to bind the Dvl PDZ domain–the peptide TMEM88-C and the small-molecule inhibitor 3289–8625 (*Grandy et al., 2009*); we reasoned that by competing with the intrinsic binding of the C-terminus, these two agents should induce an open Dvl conformation. Indeed, the two molecules enhanced JNK activation by wild-type Dvl. Interestingly, consistent with these findings, several groups have reported that JNK is activated by sulindac, a nonsteroidal anti-inflammatory drug we previously demonstrated to bind to the Dvl PDZ domain (*Czibere et al., 2005*; *Rice et al., 2006*; *Lee et al., 2009b*; *Singh et al., 2011*). Therefore, we conclude that the open conformation of Dvl is likely to initiate JNK activation.

Dvl has been suggested to exist in activated and inactivated states within the cell and to be activated by Wnt signals (*Lee et al., 2003*; *Wharton, 2003*; *Wallingford and Habas, 2005*; *Gao and Chen, 2009*); however, the form of the 'active state' has not been determined. Our findings suggest that the activation state of Dvl is determined by its conformation, such that in the absence of Wnt ligand, Dvl adopts a closed conformation that represents its inactive state.

Wnt signaling opens the closed conformation of Dvl and thereby activates Dvl. For example, in a working model of the canonical Wnt signaling pathway, the simultaneous binding of Wnt ligand to both of its membrane-bound receptors, Fz and LRP5/6, initiates canonical Wnt signaling by causing dimerization of the two receptors. Within the cell, the close proximity of the two receptors' cytoplasmic tails triggers the formation of signalosomes (*Bilic et al., 2007*) containing Dvl and Axin; Dvl binds to Fz through its PDZ domain (*Wong et al., 2003*), thus acquiring the open conformation. This open conformation also allows the Dvl DEP domain to interact with the membrane through nonspecific charge–charge interactions (*Noordermeer et al., 1994*; *Theisen et al., 1994*; *Axelrod et al., 1998*; *Wong et al., 2003*; *Park et al., 2005*; *Wang et al., 2006*; *Schwarz-Romond et al., 2007*; *Simons et al., 2009*) that in turn promote the Fz–Dvl interaction. Axin binds LRP5/6 (*Mao et al., 2001*; *Tamai et al., 2004*), and the DIX domains of Dvl and Axin interact to further stabilize the complex (*Schwarz-Romond et al., 2007*; *Fiedler et al., 2011*). The network of interactions in the super-complexes stabilizes the signalosomes remarkably, although the individual interactions are relatively weak. Energetically, the super-complex is more stable than the closed conformation of Dvl and therefore can capture the PDZ domain of Dvl. By ousting the Dvl C-terminus from its bond with the Dvl PDZ domain, the Fz binding motif also opens the conformation of Dvl as it is captured in the signalosome.

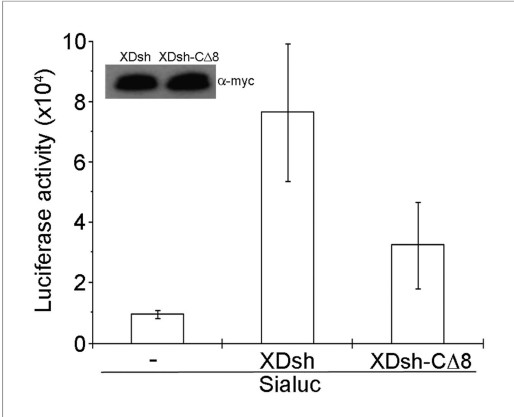

**Figure 7**. Effect of wild-type XDsh and XDsh-CΔ8 activity on canonical Wnt signaling. Luciferase assay using a *Siamois* promoter reporter (*Sialuc*). *Sialuc* DNA (200 pg) was injected alone or with myc-tagged *Xdsh* or *Xdsh-CΔ8* mRNA (500 pg) into the animal pole region of 2-cell *Xenopus* embryos. Ectodermal explants were dissected at the early gastrula stage for luciferase assay. Values are the means ±SD from four independent experiments (p < 0.05). Inset shows a representative western blot using anti-myc antibody (9E10) to control for XDsh and XDsh-CΔ8 protein expression in the four experiments.

Although other Wnt signaling pathways are less well defined, it is clear that most, if not all, non-canonical Wnt pathways are triggered by the interaction between Fz and Dvl, which is likely to open the conformation of Dvl as well (*Habas and Dawid, 2005*; *Gao and Chen, 2009*). Therefore, we propose that the active form of Dvl is its open conformation and that this active form initiates JNK-related Wnt signaling pathways.

The Dvl DEP domain is essential to activate JNK cascades (*Boutros et al., 1998*). The hypothesis that 'opened' Dvl stimulates JNK activity is consistent with a report that Daam1, a key player that connects Dvl to JNK, exists in an autoinhibited state and is activated by binding to Dvl (*Liu et al., 2008*). Daam1 binds to the Dvl DEP domain (*Habas et al., 2001*), which is likely to be obstructed in the closed conformation. Therefore, opening of Dvl's conformation is a key step in the Wnt-stimulated cascade that activates JNK.

## Materials and methods

### Protein expression and purification
The cDNAs encoding the PDZ (residues 251–340), DEP (residues 377–503), mC1 (residues 251–695), and mD1-CΔ7 (residues 251–688) domains of mouse Dvl-1 were sub-cloned into the pET28a vector. The N-terminally 6xHis-tagged proteins were expressed in BL21(DE3) *Escherichia coli* and purified by Ni-affinity chromatography followed by gel filtration chromatography as we described previously (*Wong et al., 2003*) and describe in the supplementary information.

### Peptide synthesis and purification
Peptides were synthesized by the Hartwell Center for Bioinformatics & Biotechnology at St. Jude Children's Research Hospital; they were purified by reverse-phase high-performance liquid chromatography and confirmed by MSI-MS as described in the supplementary information.

### Binding studies
For the fluorescence spectroscopy studies, a Fluorolog-3 spectrofluorometer (HORUBA Instruments Inc, Edison, NJ) with a $10 \times 4$ mm quartz cell (Hellma Inc.) with magnetic stirring was used. To confirm that the binding site of the Dvl-1 PDZ domain was occupied by intrinsic C-terminus, we generated two proteins, mC1 and mC1-CΔ7, which were separately titrated into the fluorescence-labeled peptide Rox-DprC. The $K_I$ of both peptides was determined in two independent experiments by using the equation $K_D^{app} = K_D (1 + [I]/K_I)$, where $K_D^{app}$ is the apparent $K_D$ of Rox-Dpr-C with the Dvl-C or Dsh-C peptide and [I] is the concentration of both peptides (*Figure 2*). For the ITC studies, Auto-iTC-200 (MicroCal) was used to obtain the binding affinity of Dvl-C peptide and Dvl PDZ protein in 50 mM phosphate buffer. The concentration of Dvl-C peptide in the syringe was 1.05 mM and the concentration of Dvl PDZ domain in the cell was 0.114 mM (*Figure 3*). The $K_D$ value was averaged from two independent experiments at 25°C.

### NMR spectroscopy
All NMR experiments were performed at 15°C using Bruker Avance 800-MHz spectrometers equipped with triple-resonance, 5-mm triple axis–shielded gradient probes. For titration experiments we used the Varian Unity INOVA 600 MHz spectrometer equipped with a triple-resonance, 5-mm triple-axis shielded gradient probe at 25°C.

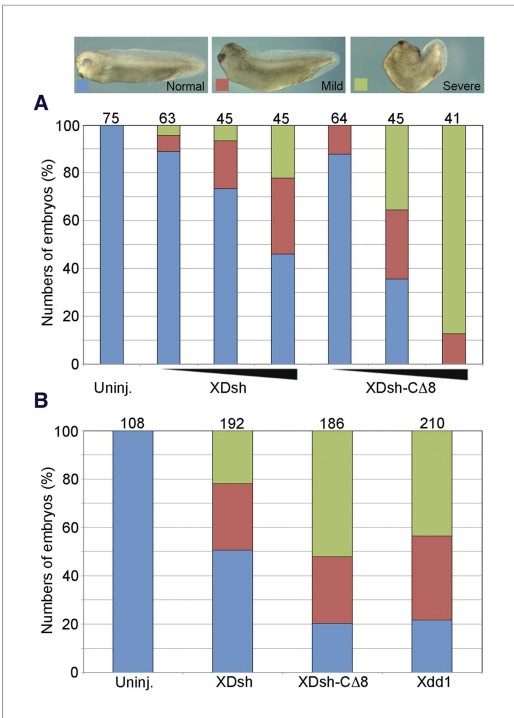

**Figure 8**. The open conformation of Dvl significantly enhances gain-of-function planar cell polarity (PCP) signaling. (**A**) Xenopus embryonic abnormal convergent extension (CE) phenotypes induced by injection of wild-type *XDsh* and *XDsh-CΔ8* mRNA at three increasing concentrations (arrows at bottom represent 80 pg, 200 pg and 500 pg of injected mRNA; above are numbers of embryos injected from two independent experiments). Phenotypes are severe (green), mild (red), and normal (blue). (**B**) Comparison of phenotypes induced by dose-equivalent injections (500 pg mRNA) of *XDsh*, *XDsh-CΔ8*, and *Xdd1* (a well-established dominant-negative *XDsh* mutant). XDsh-CΔ8 and Xdd1 induced similar phenotypes. The numbers of embryos injected from three independent experiments are listed on the top of each column.

## Structure determination of the Dvl-1 PDZ/Dvl-C peptide complex

We used NMR-derived data and a simulated annealing protocol using the program CNS within the HADDOCK software (ver. 1.2) (*Dominguez et al., 2003*). Coordinates of mouse Dvl PDZ were taken from the X-ray structure of the *Xenopus* PDZ domain (1L6O:A), whose amino acid residues were modified to fit the mouse Dvl PDZ domain (*Wong et al., 2003*). Two different types of restraints were used to determine the complex structures: (1) Ambiguous interaction restraints were chosen on the basis of the chemical shift perturbation, intermolecular NOEs between the Dvl-C peptide/PDZ complex, and solvent accessibility (calculated by using the program NACCESS [*Hubbard and Thornton, 1993*]); (2) Unambiguous distant restraints were obtained from several different types of NOESY experiments, including 2D [F1,F2]-double filtered NOESY experiments and 3D F1-half-filtered and F2-edited $^{13}$C-NOESY-HSQC experiments using the $^{13}$C/$^{15}$N PDZ domain of Dvl with unlabeled Dvl-C peptide. NOE restraints were grouped into distance ranges according to their relative intensity: strong (1.8–2.5 and 1.8–3.0 Å), medium (1.8–4.0 Å), and weak (1.8–5.0 Å). A total of 45 unambiguous restraints (23 sequential intramolecular NOEs from the Dvl-C peptide bound to the PDZ domain and 22 intermolecular NOEs between the Dvl-C peptide and the Dvl-1 PDZ domain) were used. Two types of restraints were combined to generate 2000 initial structures of the Dvl PDZ/Dvl-C peptide complex; 200 of these structures were selected by using NOE-derived restraints, and 100 structures were then obtained for final refinement. We ultimately selected the 15 lowest-energy conformers from the final 100 complex structures for further structural analysis.

## *Xenopus* embryos and microinjection of mRNA

*Xenopus* eggs were obtained from females previously injected with 500 IU of human chorionic gonadotropin (Sigma) and artificially fertilized. Synthesis, microinjection of capped mRNAs, and treatment of ectodermal explants with activin were previously described (*Carron et al., 2005*). After microinjection, some embryos were incubated in medium containing 2 µg/ml of the small-molecule compound 3289–8625 until the desired stage (*Grandy et al., 2009*).

## Luciferase-based assays

To examine canonical Wnt signaling, both the *siamois* promoter–driven luciferase reporter (*Brannon et al., 1997*) and the TOPFLASH luciferase reporter were used. The *siamois* promoter reporter DNA construct (*Sialuc*, 200 pg) or the TOPFLASH reporter DNA construct (200 pg) was injected, alone or with mRNA (500 pg) encoding wild-type XDsh or a mutant XDsh-CΔ8 lacking the PDZ-binding motif, into the animal pole region of 2-cell *Xenopus* embryos. Ectodermal explants were dissected from

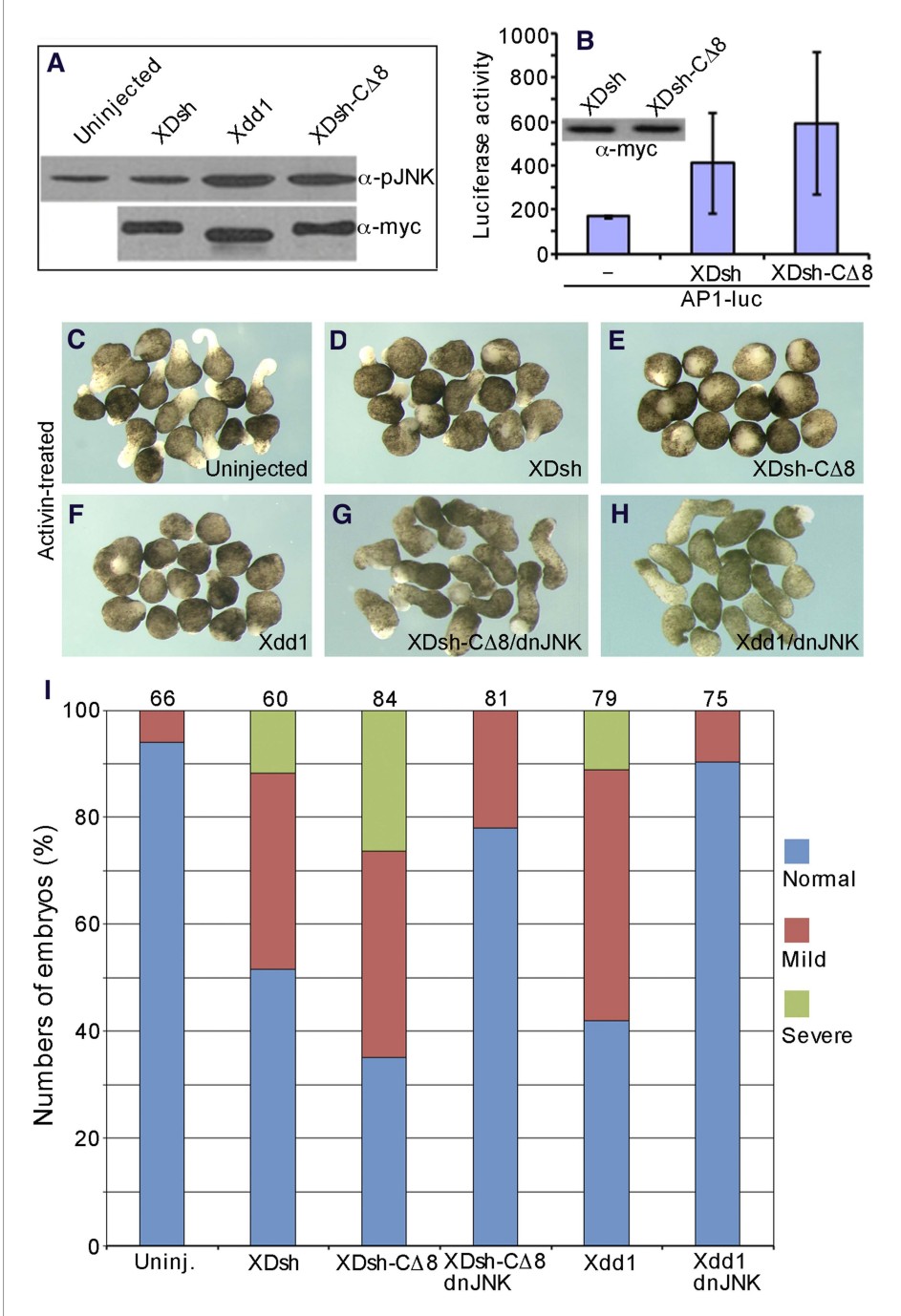

**Figure 9**. The open conformation of Dvl disrupts CE by activating Jun N-terminal kinase (JNK). (**A**) Western blot of phosphorylated JNK in ventral mesoderm cells overexpressing wild-type XDsh or its mutants. At equivalent protein level, Xdd1 and XDsh-CΔ8 more potently induce JNK phosphorylation than wild-type XDsh. (**B**) Xenopus 4-cell stage embryos were dorsally coinjected with equal quantities of wild-type *XDsh* mRNA or *XDsh-CΔ8* mRNA (500 pg) and the AP1-luciferase reporter DNA (200 pg); luciferase activity was assayed at the late gastrula stage. Inset shows a representative Western blot using anti-myc antibody to control XDsh and XDsh-CΔ8 protein levels. Values are the mean and SD from three independent experiments (XDsh vs XDsh-CΔ8, p < 0.05). (**C–H**) The dominant negative JNK mutant (dnJNK) rescues activin-induced explant elongation blocked by overexpression of XDsh-CΔ8 or Xdd1. (**C**) Uninjected explants treated with activin show extensive elongation. (**D**) *XDsh*-injected explants treated with activin show moderate inhibition of explant elongation. (**E**) Injection of *XDsh-CΔ8* strongly inhibits explant elongation. (**F**) Injection of *Xdd1* similarly inhibits explant elongation as injection of *XDsh-CΔ8*. (**G**, **H**) dnJNK rescues

*Figure 9. continued on next page*

*Figure 9. Continued*

explant elongation inhibited by XDsh-CΔ8 or Xdd1. (**I**) dnJNK also recues CE defects produced by overexpression of XDsh-CΔ8 or Xdd1 in whole embryos. Phenotypes are severe (green), mild (red), and normal (blue). Numbers on the top indicate total embryos scored from three independent experiments.

injected embryos at the late blastula stage. To assess non-canonical Wnt signaling, we used the AP1-luciferase reporter (*Rui et al., 2007*) to monitor activation of JNK in *Xenopus* whole embryos. The reporter DNA (200 pg) was injected alone or coinjected with equal quantities (500 pg) of *XDsh-CΔ8* or wild-type *XDsh* mRNA into the dorsal region of 4-cell *Xenopus* embryos, which were allowed to develop to late gastrula stage. A Lumat LB9507 luminometer (Berthold Technologies GmbH & Co) was used to perform the luciferase assays (Promega). We used cell lysates from 10 explants or five whole embryos to measure luciferase activity. All experiments were performed at least in triplicate using different batches of embryos and the mean value was calculated using Student's *t*-test.

## Western blot assays

Synthetic mRNAs (500 pg) corresponding to myc-tagged wild-type XDsh and different mutants were injected into the ventral blastomeres of *Xenopus* embryos at the 4-cell stage. At the early gastrula stage, 10 ventral mesoderm explants were dissected and analyzed by western blot using anti-phospho-JNK (Thr183/Tyr185, Thr221/Tyr223) antibody (Millipore) and anti-myc 9E10 antibody (Santa-Cruz Biotechnology).

## *Xenopus* whole embryo and ectodermal explant studies

The two dorsal blastomeres of 4-cell *Xenopus* embryos were injected with mRNA (500 pg) encoding XDsh, XDsh-CΔ8, or Xdd1, either alone or with mRNA (500 pg) encoding dnJNK. For the dose-depending studies, three different amounts of *XDsh* and *XDsh-CΔ8* mRNAs (80 pg, 200 pg and 500 pg) were used. The embryos were cultured to the larval stage, then grouped and counted according to normal, mild, or severe CE abnormal (JNK gain-of-function) phenotype. To investigate the effect of XDsh, XDsh-CΔ8, and Xdd1 on CE in vitro, the same mRNAs were injected, alone or with *dnJNK* mRNA, into the animal pole region of 2-cell *Xenopus* embryos. Ectodermal explants were dissected at the early blastula stage, incubated with activin for 1 hr, cultured to the early neurula-stage equivalent, and examined for the extent of explant elongation. To investigate how the C-terminal region of TMEM88 affects Dvl function, equal amounts of mRNAs (500 pg) encoding the C-terminal half of TMEM88 (TMEM88-C) and wild-type XDsh were coinjected into the two dorsal blastomeres

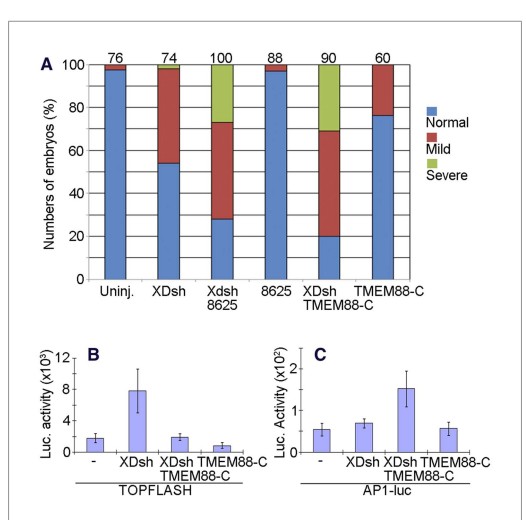

**Figure 10**. The open conformation of Dvl induced by targeting the Dvl PDZ domain potentiates Wnt/JNK signaling. (**A**) Regulation of XDsh-mediated PCP signaling by a PDZ-binding small molecule or peptide is shown by the gain-of-function CE phenotypes of whole embryos that were uninjected (controls) or injected with *XDsh* mRNA with or without treatment with the Dvl inhibitors 3209–8625 or coinjected with *XDsh* and *TMEM88-C* mRNAs. (**B**) Inhibiting the Dvl PDZ domain blocks canonical Wnt signalling induced by Dvl overexpression. Wild-type *XDsh* mRNA was injected alone or coinjected with an equal quantity of *TMEM88-C* mRNA in the animal pole region of two-cell stage Xenopus embryos, and ectodermal explants were dissected at the late blastula stage. TOPFLASH luciferase activity values are the mean and SD from three independent experiments (p < 0.05). (**C**) Inhibition of the Dvl PDZ domain by TMEM88 opens the conformation of Dvl and potentiates Wnt/JNK signaling induced by Dvl overexpression. Xenopus 4-cell stage embryos were injected dorsally with wild-type *XDsh* mRNA or coinjected with equal quantities of wild-type *XDsh* mRNA and *TMEM88-C* mRNA. AP1 luciferase activity was assayed at the late gastrula stage. Values are the mean and SD from three independent experiments (p < 0.05).

at the 4-cell stage, and the embryos were again grouped and counted as having normal, mild, or severe CE phenotypes. The above experiments were performed at least twice using different batches of embryos.

## Acknowledgements

We thank the St. Jude Hartwell Center for Bioinformatics and Biotechnology for computational time and peptide synthesis (Scott Malone and Mi Zhou for computer-related technical support and Robert Cassell and Dr Patrick Rodrigues for peptide synthesis). We thank Drs Weixing Zhang and Royappa Grace for assistance with NMR experiments; Youming Shao, Cristina Guibao, S Sokol, SC Lin and D Kimelman for plasmids; and Sharon Naron for editing the manuscript. This work was supported by NIH grants CA21765 (Cancer Center Support Grant) and GM081492, the American Lebanese Syrian Associated Charities (ALSAC) and an unrestricted grant from Research to Prevent Blindness (JJZ), and by grants from the Association Française contre les Myopathies, Ligue Nationale Contre le Cancer, Association pour la Recherche sur le Cancer, Agence Nationale de la Recherche (ANR-09-BLAN-0262-03), and the National Natural Science Foundation of China (31271556, 31471360) (DLS).

## Additional information

### Funding

| Funder | Grant reference | Author |
|---|---|---|
| National Institute of General Medical Sciences (NIGMS) | GM081492 | Jie J Zheng |
| National Cancer Institute (NCI) | CA21765 | Jie J Zheng |
| Agence Nationale de la Recherche (L' Agence Nationale de la Recherche) | ANR-09-BLAN-0262-03 | De-Li Shi |
| Research to Prevent Blindness (RPB) | JSEI | Jie J Zheng |
| National Natural Science Foundation of China | 31271556 | De-Li Shi |
| National Natural Science Foundation of China | 31471360 | De-Li Shi |

The funders had no role in study design, data collection and interpretation, or the decision to submit the work for publication.

### Author contributions

H-JL, D-LS, Acquisition of data, Analysis and interpretation of data; JJZ, Conception and design, Analysis and interpretation of data, Drafting or revising the article

### Author ORCIDs

Jie J Zheng, http://orcid.org/0000-0001-6524-6800

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
