## [Decision Letter]

Thank you for submitting your work entitled “Conformational change of Dishevelled plays a key regulatory role in the Wnt signaling pathways” for peer review at *eLife*. Your submission has been favorably evaluated by John Kuriyan (Senior editor), a Reviewing editor, and three reviewers, one of whom, Sergei Sokol, has agreed to share his identity.

The reviewers have discussed the reviews with one another and the Reviewing editor has drafted this decision to help you prepare a revised submission. The three full reviews (lightly edited) are included in this letter, as there are some specific and useful suggestions in them that will not be repeated in the summary here.

All of the reviewers were impressed with the importance and novelty of your work. We look forward to receiving the revised version of your manuscript.

Reviewer #1:

Lee et al. present evidence that the Dvl PDZ domain binds to the Dvl C-terminus, and that disrupting this interaction leads to activation of Wnt/JNK signaling. Since Dvl critical for both Wnt /β-catenin and PCP signaling, this is a very intriguing set of findings. The work seems technically solid. There are a few points that the authors should address, however.

1) The intramolecular interaction of the PDZ and C-terminus is described as a “closed conformation”. Do the authors see differences in hydrodynamic radius of the purified proteins (e.g. SEC, DLS) when the interaction is disrupted?

2) The data in Figure 7 are used to argue that the Dvl C-terminus has an important role in Wnt/β-catenin signaling, since deletion of this region doesn't produce a strong activation of luciferase activity. Yet high doses the full-length protein gives the well-known effect. Is this because some fraction of the overexpressed full-length protein is “open”? Also, what does deletion of the PDZ domain do in this assay?

3) Since deletion of either the PDZ or the C-terminus produces similar phenotypes, neither is implicated in directly activating JNK signaling. It is not clear from the Discussion how the authors think the activation may be occurring. Also, do the authors have an idea about a potential role of the PDZ-Fzd interaction?

4) I originally thought that the “closed conformation” is the one used for beta-catenin activation, but this is not what the authors are claiming – they think ablating the PDZ-C-term interaction is important for both, although they muddle through this point in the penultimate paragraph. I think we agree that they need to clarify their discussion on beta-catenin signaling. I also agree that they need to discuss the earlier papers on Dvl/JNK signaling more carefully.

5) Re Figure 2, this also confused me until I realized that the protein alone lanes have no beads – these are just input protein markers. But, they don't do the control of beads alone nor (better) a scrambled peptide. In any case this needs to be corrected.

Reviewer #2:

This work relates to the mechanism of Dishevelled activation by the Wnt pathway. The authors provide significant evidence uncovering a physical interaction between the PDZ domain of the protein and the C-terminal PDZ-binding motif. Since the two domains might be expected to interact, this result, on its own, is not a surprise, but the authors propose that this interaction is critical for the ability of Dishevelled to signal. According to their model, the intramolecular association is responsible for the closed conformation of Dishevelled, which prevents JNK activation and convergent extension defects (referred to as noncanonical signaling). The comparison of Dishevelled constructs shows that the protein with a deleted C-terminus (C∆8) and the one with a nonfunctional PDZ domain (Xdd1) activate JNK and interfere with convergent extension stronger than the wild-type protein. Similar effects are observed for two Dvl antagonists that bind to the PDZ domain, consistent with the proposed model. Interestingly, the effects on beta-catenin reporters are opposite to those obtained for AP-1 luciferase, suggesting that the ‘open conformation’ of Dishevelled inhibits canonical Wnt signaling. This is another important point, although it is not emphasized in the manuscript.

This study is interesting because of the critical role of Dishevelled in Wnt signaling, the experiments are overall solid, with some typos and controls missing. To verify their conclusions obtained for the C-terminally truncated construct (C∆8), the authors are recommended to make specific point mutations disrupting the PDZ binding motif and, possibly, the PDZ domain. They should also confirm that the inhibitors used (TMEM88-C and 8625) interfere with the PDZ-PBD binding in vitro. I feel that the paper would benefit from showing an effect of a Wnt signal on the proposed intramolecular interaction, although these experiments might be considered beyond the scope of the present study. The authors should discuss early papers on Dishevelled and JNK signaling, such as [5] in Cell, which are not fully consistent with the proposed model.

Reviewer #3:

The authors address the potential role of intramolecular interactions between the Dsh/Dvl C-terminal sequences and its PDZ domain. They establish through a series of biochemical/structural techniques and a follow up biology in *Xenopus* that Dsh/Dvl is normally in a closed confirmation (with the C-term residues bound to the PDZ domain) and open Wnt-signaling activation it is “opened” up with the C-term residues being released from the PDZ and allowing the full length protein to bind to and activate downstream effectors, like for example JNK signaling in the Wnt/PCP signaling context or interaction with Axin in the canonical signaling pathway.

The observations presented here are of significance and relevance and worth publishing in *eLife*. I have however a few concerns that should be addressed prior to publication.

1) Based on the presented data, both the canonical and the PCP (non-canonical) Wnt pathway are more “effective” in the open configuration. Yet, in the Abstract the authors state: “Wnt-regulated CE is more readily affected by Dvl mutants unable to form the closed conformation than wild-type Dvl...non-canonical signaling is mediated by the open conformation of Dvl...”. I assume the authors refer to CE and non-canonical signaling as the same (PCP-type Wnt signaling) and yet they state that one is “more affected”. This part of the Abstract is very unclear and needs to be rewritten.

2) The authors state (in the subsection “The binding pocket of Dvl PDZ is occupied by its intrinsic ligand”): “... suggesting not only that the Dvl PDZ domain binds intrinsically to the Dvl C-terminus but also that this binding interferes with interactions between the Dvl PDZ domain and its other binding partners in the Wnt signaling pathways”. How does this work with the presumed Fz C-tail Dvl PDZ interaction, which has been shown to be essential for Fz induced Dsh/Dvl membrane recruitment and thus activation; this was published by the same authors [Wong et al., Mol Cell 2003].

3) Throughout the paper the biochemical and biological experiment quantifications lack statistical analyses, e.g. Figures 7, 8, 9 and 10.

4) The data presented in Figure 9 lacks several controls. The authors need to include “uninjected” and “xDsh alone”.

5) Some biological assays in a real in vivo/genetic setting (mouse or *Drosophila*) would have been helpful. The *Xenopus* experiments suffer from massive overexpression and pure gain-of-function approaches. Does the model hold up in a loss-of-function genetic context?

---

## [Author Response]

Reviewer #1:

*Lee et al. present evidence that the Dvl PDZ domain binds to the Dvl C-terminus, and that disrupting this interaction leads to activation of Wnt/JNK signaling. Since Dvl critical for both Wnt /β-catenin and PCP signaling, this is a very intriguing set of findings. The work seems technically solid. There are a few points that the authors should address, however*.

1) The intramolecular interaction of the PDZ and C-terminus is described as a “closed conformation”. Do the authors see differences in hydrodynamic radius of the purified proteins (e.g. SEC, DLS) when the interaction is disrupted?

We did not see definitive difference between “closed” and “open” forms of Dvl in the SEC studies. Our DLS studies were also unsuccessful partially because it was very difficult to concentrate the proteins. However, we did obtain a set of data from an analytical ultracentrifugation study (shown in Figure 11). The shapes of the data fit perfectly well with our model; the data shows that the “closed” form (C1) is more globular than the “open” form (C1-Δ7) (Figure 11). However, the numbers extracted from data, such as hydrodynamic radius (Rstocks), for the two forms of Dvl are essentially the same. We speculated that this might be due to the large flexible loop in the “closed” form. Because of this inconsistence, we decided not to include the ultracentrifugation data in the manuscript.

Author response image 1.**DOI:**
http://dx.doi.org/10.7554/eLife.08142.018

*2) The data in*
Figure 7
*are used to argue that the Dvl C-terminus has an important role in Wnt/β-catenin signaling, since deletion of this region doesn't produce a strong activation of luciferase activity. Yet high doses the full-length protein gives the well-known effect. Is this because some fraction of the overexpressed full-length protein is “open”? Also, what does deletion of the PDZ domain do in this assay?*

Indeed, some fraction of the full-length Dvl is in “open” form. This issue is discussed and showed in Figure 6 in the manuscript.

Because the Fzd-PDZ interaction plays a key role in Wnt/β‐catenin signaling, therefore, deletion of the PDZ domain will block the signaling. In fact, it was showed that deletion of the PDZ domain abolishes the activity of Dsh in canonical Wnt signaling ([41], Curr Biol 6:1456‐1467). However, it is a different issue from what we aim to address in the current manuscript.

3) Since deletion of either the PDZ or the C-terminus produces similar phenotypes, neither is implicated in directly activating JNK signaling. It is not clear from the Discussion how the authors think the activation may be occurring. Also, do the authors have an idea about a potential role of the PDZ-Fzd interaction?

As we discussed in the Discussion section, we clearly state that although we propose a few possibilities, we don’t know exactly how JNK is activated through Dvl. What we report in the manuscript is our observations that the “open” conformation of Dvl was more effective in triggering JNK activation.

Regarding the potential role of the PDZ-Fzd interaction, as we discuss in the Discussion section, this interaction should open the “closed” Dvl to the “open” form.

*4) I originally thought that the “closed conformation” is the one used for beta-catenin activation, but this is not what the authors are claiming – they think ablating the PDZ-C-term interaction is important for both, although they muddle through this point in the penultimate paragraph. I think we agree that they need to clarify their discussion on beta-catenin signaling. I also agree that they need to discuss the earlier papers on Dvl/JNK signaling more carefully*.

Actually, at this moment, based on all the data we have, we believe that the “open” conformation of Dvl is more active in all the Dvl mediated Wnt signaling pathways. This issue is discussed in the Discussion section.

*5) Re*
Figure 2*, this also confused me until I realized that the protein alone lanes have no beads – these are just input protein markers. But, they don't do the control of beads alone nor (better) a scrambled peptide. In any case this needs to be corrected*.

This point is well taken; we removed the figure in the revised manuscript.

Reviewer #2:

*[…] This study is interesting because of the critical role of Dishevelled in Wnt signaling, the experiments are overall solid, with some typos and controls missing. To verify their conclusions obtained for the C-terminally truncated construct (C∆8), the authors are recommended to make specific point mutations disrupting the PDZ binding motif and, possibly, the PDZ domain*.

Because the nature of PDZ domain, it is very difficult to identify such a mutant that has significant reduction in binding affinity to its binding partners yet maintain the integrity of the protein.

*They should also confirm that the inhibitors used (TMEM88-C and 8625) interfere with the PDZ-PBD binding in vitro*.

We did the experiment according to the reviewer’s suggestion and the data are incorporated in the new Figure 10.

*I feel that the paper would benefit from showing an effect of a Wnt signal on the proposed intramolecular interaction, although these experiments might be considered beyond the scope of the present study*.

We discuss this issue in the Discussion section.

*The authors should discuss early papers on Dishevelled and JNK signaling, such as*
[5]
*in Cell, which are not fully consistent with the proposed model*.

The reference was mistakenly omitted during the editing of the original manuscript. We sincerely apologize for this mistake. The reference is added back.

We think our proposed model is actually consistent with the Cell paper. This issue is discussed in the Discussion section.

Reviewer #3:

*[…] The observations presented here are of significance and relevance and worth publishing in* eLife*. I have however a few concerns that should be addressed prior to publication*.

*1) Based on the presented data, both the canonical and the PCP (non-canonical) Wnt pathway are more “effective” in the open configuration. Yet, in the Abstract the authors state: “Wnt-regulated CE is more readily affected by Dvl mutants unable to form the closed conformation than wild-type Dvl...non-canonical signaling is mediated by the open conformation of Dvl...”. I assume the authors refer to CE and non-canonical signaling as the same (PCP-type Wnt signaling) and yet they state that one is “more affected”. This part of the Abstract is very unclear and needs to be rewritten*.

We reword the sentence in the Abstract according to the suggestion of the reviewer.

*2) The authors state (in the subsection “The binding pocket of Dvl PDZ is occupied by its intrinsic ligand”): “... suggesting not only that the Dvl PDZ domain binds intrinsically to the Dvl C-terminus but also that this binding interferes with interactions between the Dvl PDZ domain and its other binding partners in the Wnt signaling pathways”. How does this work with the presumed Fz C-tail Dvl PDZ interaction, which has been shown to be essential for Fz induced Dsh/Dvl membrane recruitment and thus activation; this was published by the same authors [Wong et al., Mol Cell 2003]*.

Indeed, the Dvl PDZ domain can bind to Dvl C-terminus, Fz C-terminal tail, and many other binding partners. We propose that inside the cell, these interactions compete with each other, and the surrounding condition the Dvl inside the cell determines the binding partner(s) of the Dvl PDZ domain. We discussed this issue in the Discussion section. For example, in the current working model of the canonical Wnt signaling, the binding of Wnt to both its receptors, Fz and LRP5/6 triggers the formation of the signalosome. The network of interactions in the signalosome super-complexes stabilizes the signalosomes remarkably, although the individual interactions are relatively weak. Energetically, the supercomplex is more stable than the “closed” conformation of Dvl and therefore can capture the PDZ domain of Dvl. By ousting the Dvl C-terminus from its bond with the Dvl PDZ domain, the Fz binding motif also opens the conformation of Dvl as it is captured in the signalosome.

*3) Throughout the paper the biochemical and biological experiment quantifications lack statistical analyses, e.g.*
Figures 7, 8, 9 and 10.

We now include statistical analyses in these figure legends. Particularly, for luciferase assays, Student t-test was used in our analysis and the P values are included. For whole embryo phenotypes, at least two independent experiments were performed, with total numbers of embryos being indicated.

*4) The data presented in*
Figure 9
*lacks several controls. The authors need to include “uninjected” and “xDsh alone”*.

These injections are included in Figure 9.

*5) Some biological assays in a real in vivo/genetic setting (mouse or* Drosophila*) would have been helpful. The* Xenopus *experiments suffer from massive overexpression and pure gain-of-function approaches. Does the model hold up in a loss-of-function genetic context?*

We truly believe that the model we proposed should hold up in different loss-of-function genetic contexts. However, we will not be able to perform these studies in a very limited time frame and we also think that the experiments are out of the scope of this manuscript. Nevertheless, we plan to carry out some of the studies along those lines in the future.